# Contextualizing gender disparities in online teaching evaluations for professors

**Xiang Zheng**[1], **Shreyas Vastrad**[1], **Jibo He**[2], **Chaoqun Ni**[1]*

**1** Information School, University of Wisconsin-Madison, Madison, Wisconsin, United States of America,
**2** Department of Psychology, Tsinghua University, Beijing, China

* chaoqun.ni@wisc.edu

## Abstract

Student evaluation of teaching (SET) is widely used to assess teaching effectiveness in higher education and can significantly influence professors' career outcomes. Although earlier evidence suggests SET may suffer from biases due to the gender of professors, there is a lack of large-scale examination to understand how and why gender disparities occur in SET. This study aims to address this gap in SET by analyzing approximately 9 million SET reviews from RateMyProfessors.com under the theoretical frameworks of role congruity theory and shifting standards theory. Our multiple linear regression analysis of the SET numerical ratings confirms that women professors are generally rated lower than men in many fields. Using the Dunning log-likelihood test, we show that words used in student comments vary by the gender of professors. We then use BERTopic to extract the most frequent topics from one- and five-star reviews. Our regression analysis based on the topics reveals that the probabilities of specific topics appearing in SET comments are significantly associated with professors' genders, which aligns with gender role expectations. Furtherly, sentiment analysis indicates that women professors' comments are more positively or negatively polarized than men's across most extracted topics, suggesting students' evaluative standards are subject to professors' gender. These findings contextualize the gender gap in SET ratings and caution the usage of SET in related decision-making to avoid potential systematic biases towards women professors.

## Introduction

Student evaluation of teaching (SET) plays a significant role in the career outcomes of instructors in higher education [1]. Many colleges and universities use SET results in decision-making regarding tenure, promotion, compensation, hiring, and other career opportunities [2–4]. SET may affect professors' self-efficacy in teaching and students' future course enrollment choices [5]. Although teacher expertise, teaching, and academic leadership are the main related factors of SET results [6], the results are not immune to biases due to teaching-irrelevant factors such as a professor's gender [7–14]. Gender bias in SET occurs if a professor's gender affects teaching evaluations provided by students, either positively or negatively, and is unrelated to the criteria of sound teaching [15]. There has been increasing attention to gender disparities in SET

**Data Availability Statement:** Minimal underlying datasets, web scraping scripts, and analytical codes are available on GitHub (https://github.com/UWMadisonMetaScience/rmp). Human names and comments are removed from the minimal

underlying datasets due to ethical considerations. The complete dataset can be collected from RateMyProfessors.com.

**Funding:** Dr. Jibo He is supported by the National Key R&D Program of China (2022YFB4500600). Dr. Chaoqun Ni is supported by the Wisconsin Alumni Research Foundation of University of Wisconsin-Madison (135-AAI3865).

**Competing interests:** The authors have declared that no competing interests exist.

ratings, some of which call for contextualizing the gender gaps under socio-cultural norms [16,17]. Given the critical role of SET in assessing the career outcomes of professors, understanding how and why SET disadvantages women professors should be essential to addressing this issue from the source and improving gender equality in higher education and academia.

However, the inaccessibility of SET data has been limiting large-scale examinations of SET. Universities and colleges typically collect SET records from students at the end of a semester and keep them confidential only for making internal personnel decisions [18]. The rise of online teaching evaluation platforms in the past decade changed the game by enabling large-scale, far-reaching, and transparent SET for analysis, among which RateMyProfessors.com (RMP) is one of the most popular platforms. RMP is an online higher education SET website allowing users to post anonymous and publicly open instructor reviews, mainly covering colleges and universities in the United States and Canada [19]. RMP allows users to rate professors on a scale of 1 to 5 and write open-ended comments. According to RMP, its users have added more than 19 million ratings, 1.7 million instructors, and 7,500 schools to its review database as of 2022 [20]. The platform has transformed the SET landscape and given students a voice in evaluating pedagogy. RMP provides a virtual space for anonymous, publicly accessible reviews and extends public consideration of students' perspectives, regardless of their socio-economic background [1].

Exploiting the large-scale SET reviews from RMP can provide insight into how students evaluate their professors in traditional, closed SETs. Many studies have found a strong correlation between RMP reviews and traditional SET results [2,21,22]. Although not adopted directly, RMP reviews can be related to decision-making concerning professors' career outcomes, such as promotions, tenure, and financial rewards [2,23,24]. In the meantime, like traditional SET, RMP reviews have faced criticism for not accurately reflecting teaching effectiveness [25,26] and for having biases, including gender bias [27]. Given its popularity and comprehensive coverage, studying the potential gender bias on RMP would shed light on understanding the context of gender biases in SET with more robust evidence.

Moreover, studying the gender biases on RMP is necessary, given its profound social impact. Maintaining RMP as a reliable and bias-free platform for measuring teaching effectiveness is critical for both students and professors. Students tend to believe that RMP is accurate [26,28] and are influenced by RMP no less than traditional SET [5]. Students may decide on their course enrollment based on RMP reviews [29], which can be more impactful than in-person discussions with friends [30]. RMP evaluations also affect students' learning motivation, self-efficacy, and perception of the professors' credibility and attractiveness [31,32]. Thus, any gender bias on RMP may discourage students' enrollment in specific courses taught by professors of a particular gender and stigmatize these professors' reputations [33]. As RMP reviews are often harsher on professors than traditional SET and are publicly available, professors who are more likely to face biases in SET results may be even more pressured by RMP [12,34,35].

This study aims to analyze and contextualize potential gender disparities in SET in US higher education using approximately 9 million reviews from RMP. We systematically compared SET ratings between men and women professors across fields. The context of the disparities was also investigated by comparing the word usage, topics, and sentiments in comments for professors of different genders. We used the role congruity theory and shifting standards theory as the theoretical framework to help design our study and interpret our results. This study is unique in its utilization of a massive dataset across disciplines and institutions and its focus on contextualizing SET rating gender disparities using text analysis, which has received less attention in previous studies. Our findings contribute to a comprehensive and in-depth understanding of gender disparities in SET and its contexts, providing implications for addressing science and workforce inequality. It is noted that although we fully agree that

"instructor" and "professor" may refer to a slightly different group of people in higher education, we use the two titles interchangeably in the study, given the term that RMP provided.

## Related research and theoretical framework

### Gender disparities in academia and teaching evaluation

Academia has had a hard time keeping women. Women professors in the US higher education are underrepresented across all professorship ranks [36] and often face glass ceilings in their career development that block them from promotion [37]. Reportedly, women professors are less likely to receive project funding [38,39] or prestigious awards than men [40,41]. Women are also found to receive fewer citations and lower salaries for their research [42,43]. While researchers suggested various factors contributing to the observed gender disparities in academia, including unbalanced research activities [42], gender gaps in parenthood [44], and unfair credit distribution [45], teaching may also come into play. Women professors often take on heavier teaching loads than men, leaving less time for research [46]. Negative feedback and comments can also decrease professors' self-efficacy [5], and women are reportedly more negatively impacted than men [18]. SET favors men and has more negative ratings and comments towards women, although the effect may be minor in some cases [4,10,11]. This gender bias may increase women's working and mental pressure, leading to their attrition from the academic pipeline [11]. Consequently, any gender biases present in SET may harm women professors who are already underrepresented and exacerbate the gender inequalities in academia. However, substantial evidence from large-scale examinations is still lacking to confirm these claims and provide a comprehensive understanding of gender biases in SET.

A limited number of studies have explored the gender disparities in SET through analysis of RMP reviews for professors. The overall teaching quality rating, which evaluates teaching effectiveness, is the most critical RMP indicator to measure professors' performance. Nevertheless, studies found that men professors receive higher ratings than women professors in some fields [17,25,47] and across fields when controlling for other factors [24]. Some studies based on local institutions or specific fields suggest little or no gender difference in RMP reviews for professors [27,48,49], indicating such gender disparities may depend on local contexts [24]. The language used in RMP review comments is also gendered [50]. A two-professor experiment shows that RMP reviews describe women's physical appearance and personability more often than men's [12]. Students are also more likely to use "genius" or "brilliant" in their comments to describe men than women professors [51]. A topics analysis on RMP comments for institutions in Ontario, Canada, shows that men are more likely to receive comments concerning positive teaching and expectation [52].

Overall, little research has thoroughly examined gender disparities in RMP reviews across institutions and fields. The large amount of information in RMP comments has yet to be analyzed to uncover the contexts of gender disparities in SET. Here we propose our hypothesis regarding the potential gender gaps in RMP ratings across fields:

*H1: Women and men professors received different ratings in RMP across fields.*

### Theoretical framework

**Role congruity theory.** The role congruity theory by Eagly and Karau [53] might be related to the gender disparities in teaching evaluations of professors. This theory assumes that women professors will receive higher ratings if they align with the feminine gender roles. At the same time, violations may lead to negative reviews and backlash, as shown in SET studies

[33,47]. Common gender roles and stereotypes expect women to be warm, caring, friendly, emotional, and submissive, while men are expected to be smart, knowledgeable, and leading [7,33,47]. Basow et al.'s [54] suggest that the best women instructors were more likely than the best men instructors to be described as having high interpersonal skills, while the best men instructors were more likely to be described as knowledgeable. These gender roles can further lead to the stereotypes that women are less likely to achieve high performance and act as leaders [51,53]. Therefore, the gender disparities in RMP ratings may stem from students' different gender role expectations for women and men professors. We analyzed the comments in RMP reviews for word usage and topics to examine whether students' comments reflect such expectation gaps due to the professor's gender. Our hypotheses based on the role congruity theory are as follows:

*H2: RMP review comments on women and men professors have different word usage patterns reflecting students' gender role expectations.*

*H3: RMP review comments on women and men professors focus on different topics reflecting students' gender role expectations.*

**Shifting standards theory.** The shifting standards theory may also help explain gender disparities in SET [55]. It posits that evaluations are often constrained by the subjectivity of judgment and language and are based on fluid evaluative standards, which stereotypes of the target group can easily influence. For example, a positive comment for a woman leader may not carry the same weight as a positive comment for a man, as stereotypes suggest men are better leaders than women. Prior research has shown that shifting standards theory can provide insights into the gendered patterns in SET results [16]. Because SET is a nonzero-sum behavior that does not demand competition over limited resources, evaluative standards are more likely to shift between groups [55]. Although RMP tries to formulate a common-rule scale for its rating [56], students may still have subjective standards when writing reviews and contribute to gender disparities [52]. We thus make the following hypothesis:

*H4: RMP review comments show shifting evaluation standards between professors of different gender groups.*

## Methods

### Data and procedures

This study used about 9 million publicly available RMP reviews from He et al. [19]. We relied on five attributes in each review record for subsequent analyses: professor name, institution, department, quality rating, and comment. A professor may be affiliated with multiple institutions, and multiple professors may share the same name. We identified individual professors based on a unique combination of professor name, institution name, and department name. While a professor may be associated with multiple institutions, we decided to keep those professors affiliated with multiple institutions as separate individuals in our analysis because student reviews may vary by institution, department, and field. The dataset's geographical and temporal distributions of reviews and professors is in He et al. [19]. We removed all blank or non-English comments before the analysis using the python package PYCLD2 [57].

We used the research pipeline illustrated in **Fig 1** to test our hypotheses. The RMP dataset contains 9,543,998 valid reviews for 919,750 professors from 1999 to 2018. We assigned the gender categories for professors based on a binary gender classification approach developed upon specific rules and methods (see **Gender classification**). The field of professors is also a

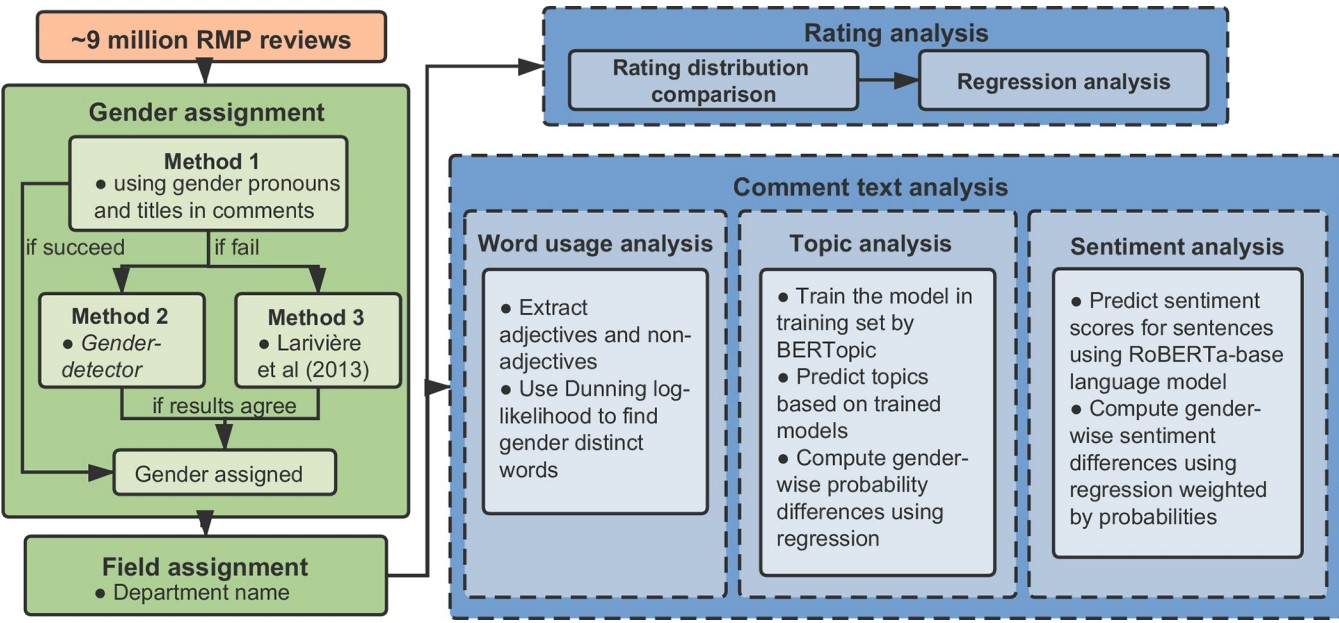

**Fig 1. Overview of research design and procedures.**

factor of our investigation, which we assigned based on the department name of individual professors (see **Field assignment**). We used the gender and field-identified data to run the rating analysis to examine if gender disparities exist for professors regarding the RMP ratings. In the comment text analysis, we focused on 4,365,780 comments with five- (the highest) or one-star (the lowest) ratings for 772,472 professors, including 428,743 (55.5%) women and 343,729 (44.5%) men, across all fields. Details are available in **Empirical analysis**.

With data privacy and ethics in mind, we de-identified individual professors for all analyses in the manuscript after gender imputation, which relied on professors' first names. Moreover, we analyzed the data at the level of gender categories instead of individuals, imposing minimum risk to human subjects. Additionally, RMP is publicly available. Per the guidelines provided by the International Review Board at the University of Wisconsin-Madison, this study is eligible for exemption from institutional review board reviews (45 CFR § 46.104 (d) (1–8)).

## Gender categorization

We inferred the gender category of professors using the following three methods. Method 1 infers gender based on gender pronouns and titles (e.g., she, her, he, his, himself, herself, hers, Ms., Miss., Mr., and Mrs.) extracted from comments. In the case of pronounces indicating contradictory gender categories for an individual professor, we used a three-to-one rule: A professor would be inferred as a man if the number of feminine pronouns is zero or lower than 1/3 of that number for masculine pronouns, and vice versa.

For names without gender classifications after Method 1, we checked if the results from Method 2 and Method 3 matched. Method 2 imputes gender categories by professor names using an open-source Python gender detection package [58]. Method 3 applies the gender classification algorithm by Larivière et al. [42], which estimates researchers' gender from their names and affiliation countries by drawing the US census data and country-specific gender-name lists. If the results from Method 2 and Method 3 matched agreed, we assigned the agreed gender to an individual. If not, we labeled the gender of professors as "unknown." We

**Table 1. The distribution of professors and share of women by field.**

| Field | All reviews | | Five-star reviews | | One-star reviews | |
|---|---|---|---|---|---|---|
| | # professors | % women | # professors | % women | # professors | % women |
| Applied Sciences | 84,830 | 35.8% | 63,092 | 35.4% | 35,731 | 36.6% |
| Natural Sciences | 97,650 | 36.8% | 73,678 | 36.3% | 45,339 | 37.3% |
| Math & Computing | 106,922 | 34.6% | 80,585 | 35.0% | 52,753 | 33.9% |
| Engineering | 26,555 | 14.2% | 18,454 | 13.4% | 11,564 | 14.4% |
| Medicine Health | 48,995 | 66.2% | 36,538 | 64.5% | 15,036 | 69.9% |
| Social Sciences | 190,409 | 46.2% | 150,986 | 45.6% | 74,252 | 45.8% |
| Education | 33,281 | 67.3% | 25,079 | 66.2% | 10,783 | 69.4% |
| Humanities | 292,071 | 49.4% | 235,049 | 48.4% | 108,845 | 50.6% |
| All | 880,713 | 44.7% | 683,461 | 44.3% | 354,303 | 44.2% |

successfully assigned gender categories to 885,555 (96.3%) professors for the entire dataset, including 687,075 professors (99.6%) with five-star reviews and 355,751 professors (99.5%) with one-star reviews. About 44.7% of the professors are women.

## Field classification

Considering teaching practices may vary by fields and departments, we grouped professors into eight broad fields based on the information of the affiliated departments [59]: Applied Sciences, Natural Sciences, Math & Computing, Engineering, Medicine Health, Social Sciences, Education, and Humanities (see **S1 Table** for the mapping between fields and departments). Professors (and corresponding reviews) affiliated with departments with unspecified names or of multi-disciplinary nature were categorized as others and excluded from the analysis. Among gender-identified professors, 880,713 professors (99.5%), including 683,461 professors (99.5%) in the five-star review subset and 354,303 professors (99.6%) in the one-star review subset, have been successfully assigned to one of the eight fields (see **Table 1**). These 880,713 professors received 8,960,812 reviews, accounting for 93.9% of reviews in the entire dataset.

## Empirical analysis

**Regression for rating analysis.** We compared the distributions and means of the ratings between women and men across fields. We then fitted a multiple linear regression model to investigate the association between gender and the overall ratings of professors. The regression model controlled for difficulty level rating [60] and fixed effects [52]. The model specification is

$$Rating_{ijk} = \beta_0 + \beta_1 Women_i + \beta_2 Difficulty_i + Year_j + Univ_k + \epsilon_{ijk} \tag{1}$$

where $Rating_{ijk}$, $Difficulty_i$, and $Women_i$ are review $i's$ overall quality rating, difficulty rating, and whether the professor was identified as a woman (yes = 1). $Year_j$ and $Univ_k$ are the review year and affiliation-level fixed effects. $\epsilon_{ijk}$ is the residual. The standard errors were clustered at the professor level.

**Word usage analysis for comments.** To compare the word usage patterns between comments for women and men professors, we adopted the Dunning log-likelihood analysis, which calculates how distinct a word is in one text corpus compared to another [61,62]. We preprocessed the data by extracting adjectives and non-adjectives in comments using spaCy [63] before applying the Dunning test. We detected whether a negation modifier modified a word (e.g., not, no) and added "not_" before the adjective if so. We manually converted common

adjectives in comments with prefixes in negation into the "not_" form. For example, "not orga-nized," "unorganized," and "disorganized" were all converted into "not_organized." Then, we computed a log-likelihood value (Dunning score) regarding the frequency of a word occurring in the two sets and the expected frequency of that word if the two sets are homogeneous. For-mally, the Dunning score for a word $w$ was computed by

$$LL_w = 2\sum_{i=1}^{2} O_{iw} \ln \frac{O_{iw}}{E_{iw}} \qquad (2)$$

where $LL$ is the Dunning score for $w$, $O_i$ is the observed counts of $w$ in the $i$th comment set, and $E_i$ is the expected counts of $w$ in the $i$th comment set, computed by

$$E_{iw} = N_i \frac{\sum_{i=1}^{2} O_{iw}}{\sum_{i=1}^{2} N_i} \qquad (3)$$

where $N_i$ is the total number of words in the $i$th comment set [61]. A high Dunning score indi-cates a large discrepancy in the word frequency between the two comment sets. We compared the usage of adjectives and non-adjectives separately between women and men professors, considering that adjectives are usually the primary source of subjective content in texts [64]. Person names, pronouns, and stop words were removed from the results.

**Topic analysis for comments.** Topic modeling is a widely used quantitative method to identify fine-grained topics from extensive textual data. Among the topic modeling techniques, BERTopic is one of the most state-of-the-art techniques using the bidirectional encoder repre-sentations from transformers (BERT) [65]. The BERTopic process transforms sentences into sentence embeddings through the Sentence-BERT framework and a pre-trained language model. The dimensionality of the embeddings is then reduced using uniform manifold approximation and projection (UMAP) [66]. The final step involves clustering the embeddings through hierarchical density-based spatial clustering of applications with noise (HDBSCAN) [67,68]. Unlike traditional methods, HDBSCAN uses a soft clustering approach, which assigns sentences a set of probabilities indicating their likelihood of belonging to a particular cluster [67,68]. BERTopic also employs the c-TF-IDF method [65] to extract keywords from the sen-tences, facilitating the interpretation and labeling of topics.

We performed topic analysis using BERTopic at the sentence level for reviews, recognizing that each review may encompass various aspects of teaching and differing emotional valences within different sentences [26,69]. Segmenting comments into sentences is also recommended for using BERTopic to identify multiple topics [70]. Sentences containing less than two non-stop words were excluded from the analysis. We masked the common gender pronouns and names with gender-neutral equivalents (e.g., he–they, Mike—Person) to reduce the potential impact of gender pronouns and person names on the clustering results. Stop words were kept before embedding transformation, as removing them may erase crucial contextual information for word embeddings and negatively impact topic extraction [71]. In accordance with the sam-ple-split workflow for text causal inference described by Egami [72], sentences were divided into training and test sets to avoid overfitting and identification problems. The training set was created through stratified random sampling, proportional to the number of reviews received by each professor. Our topic models were built and trained on two training sets, each consisting of one million sentences, for the five-star and one-star review subsets, respectively. The trained models allowed us to identify 73 topics from five-star reviews and 89 from one-star reviews.

Using the trained topic models, we predicted the probabilities of each sentence in the test sets (1,915,072 sentences in one-star reviews and 8,690,966 sentences in five-star reviews) to

**Table 2. Dimensions and coverage of topics.**

| Category | Dimension | Coverage |
|---|---|---|
| Overall | Overall | General comments on the professor or the class. |
| Professor | Teaching | Comments regarding the professor's teaching, such as how they teach the class, communicate with students, teaching effectiveness, clarity, etc. |
| | Personal | Comments regarding the professor's personal aspects that are not teaching-related, such as look, family, accent, origin, etc. |
| Course | Material | Comments regarding the materials used for the course, such as lecture notes, reading, syllabus, book, handout, study guide, lab material, manual, software, etc. |
| | Structure | Comments about the overall structure of the course, such as course design, difficulty levels, modality, etc. |
| | Evaluation | Comments regarding the evaluation of students and their performance in the course, such as assignments, projects, tests, exams, presentations, attendance, etc. |
| | Grading | Comments regarding the grading process, practice, and results |
| | Subject | Comments on the subject area of a course |
| Other/not specified | | Comments that cannot be categorized into the above categories are irrelevant to the class and professor or have no concrete meanings |

contain the identified topics. We extracted a list of the most frequent unigrams and bigrams as keywords to represent the identified topics based on the c-TF-IDF scores. To improve interpretability, we lemmatized the keywords and removed stop words from the extracted lists [70]. We selected the top three keywords as topic labels. We reviewed, evaluated, and categorized the topics by inspecting the keywords and representative sentences (see **S2 Table**). To make the results easier to interpret, we categorized the topics into nine dimensions of teaching evaluation (adapted from Brockx et al. [73]; see **Table 2**). **Fig 2** displays the distribution of comments across these dimensions.

Gender's effect on professors' probability of receiving comments on a specific topic is estimated, controlling for the comment's posting year and the professor's affiliation and field. The

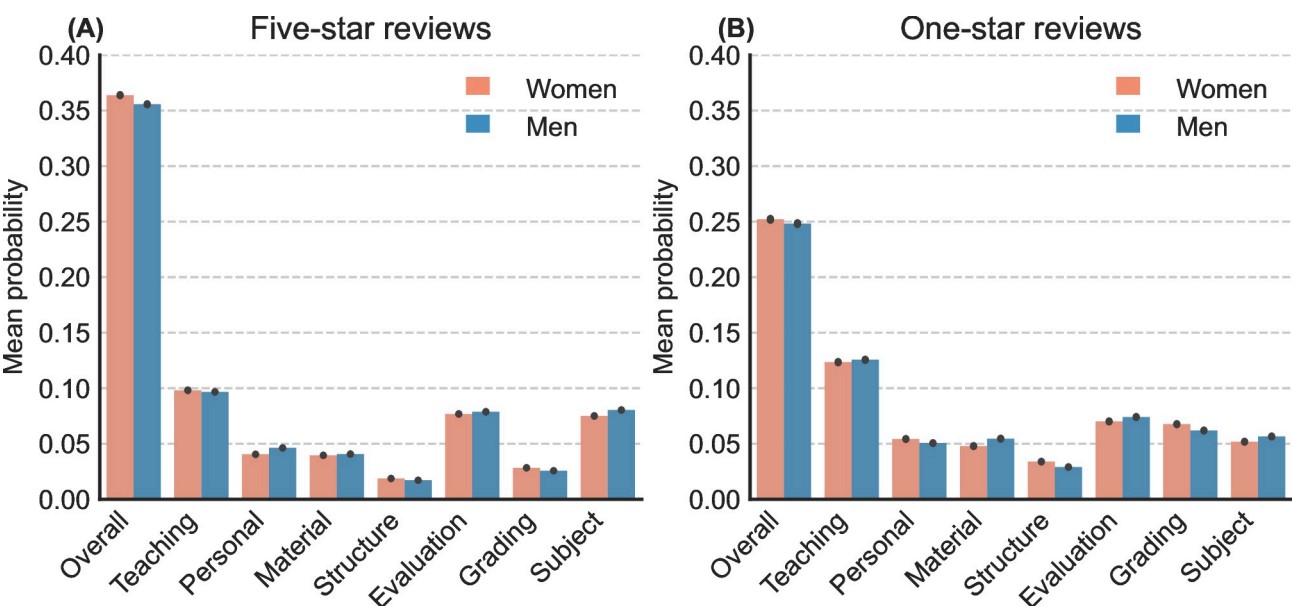

**Fig 2. Mean possibilities of comment sentences pertaining to topic dimensions by genders.** (A) In five-star reviews. (B) In one-star reviews.

following specification is used.

$$Prob_{ijkl} = \beta_0 + \beta_1 Women_i + Year_j + Univ_k + Field_l + \epsilon_{ijkl} \qquad (4)$$

where $Prob_i$ and $Women_i$ are sentence $i$'s probability for that topic and whether the professor was identified as a woman (yes = 1). $Year_j$, $Univ_k$, and $Field_l$ are the review year, affiliation, and field-level fixed effects. $\epsilon_{ijkl}$ is the residual. The standard errors were clustered at the professor level.

**Sentiment analysis for comments.** We utilized an advanced RoBERTa-base language model [74] to predict the sentiment scores of the comment sentences that were segmented during the topic analysis within the test set. The model was trained based on the original RoBERTa-base model with massive Twitter posts for masked language modeling. The model was built on 124 million tweets and fine-tuned for sentiment analysis with the TweetEval benchmark. The performance of the language model is generally competitive in predicting sentiments for various corpus data [74]. The sentiment analysis predicted three sentiment scores (positive, neutral, and negative) ranging from 0 to 1 for each sentence. We then estimated the effect of gender on the sentiment scores of sentences within a specific topic through weighted multiple linear regression. In this regression, each sentence was weighted by its probability of containing that topic, as calculated in the previous step. This means that sentences that are more likely to contain the topic carry more weight in this estimation. Like the previous regression models, this regression also included review year, affiliation, and field-level fixed effects and clustered the standard errors at the professor level.

**Data validation and sample representativeness.** According to the American Association of University Professors' report, women account for about 45.9% of total full-time faculty in the US [75]. In this study, women represent 44.7% of the professors with known gender. Among the professors with five-star and one-star reviews, 44.3% and 44.2% are women, respectively (see **Table 1**). Therefore, we conclude that the gender distribution of professors in our dataset is comparable to that of full-time faculty in the US. Additionally, we evaluated the representation of US higher education institutions in our dataset from RMP. According to the National Center for Education Statistics, there were 4,313 higher education institutions in the US in 2017–2018 [76]. Our dataset covers 4,214 institutions across all 50 states, demonstrating that RMP comprehensively covers US higher education institutions.

Additionally, we performed a data balance analysis to assess the potential impact of data skewness on our results. First, we examined the distribution of reviews received by professors of different genders to determine if certain professors in any gender group receive disproportionately more reviews. The result shows no significant disparity in the number of reviews received by women and men professors (see **Fig 3**). Second, we compared the word count of comments for professors of different genders to check for any text length differences that could affect topic analysis and sentiment prediction [77]. On average, comments for women are slightly longer than those for men in both five-star and one-star reviews (see **Table 3**). Nonetheless, subsequent analysis of comment text should be unlikely to be highly skewed by text length due to the small difference in word count (0–2 words). Furthermore, as previously mentioned, sentences containing less than two non-stop words and blank comments were excluded from the topic modeling and sentiment analysis components, ensuring that these analyses were based on sufficient and relevant text information.

## Results

### Gender disparities in ratings

From the rating distributions for professors across fields, we discovered that a large proportion of reviews tend to provide extreme ratings (see **Fig 4**). One-star and five-star reviews, which

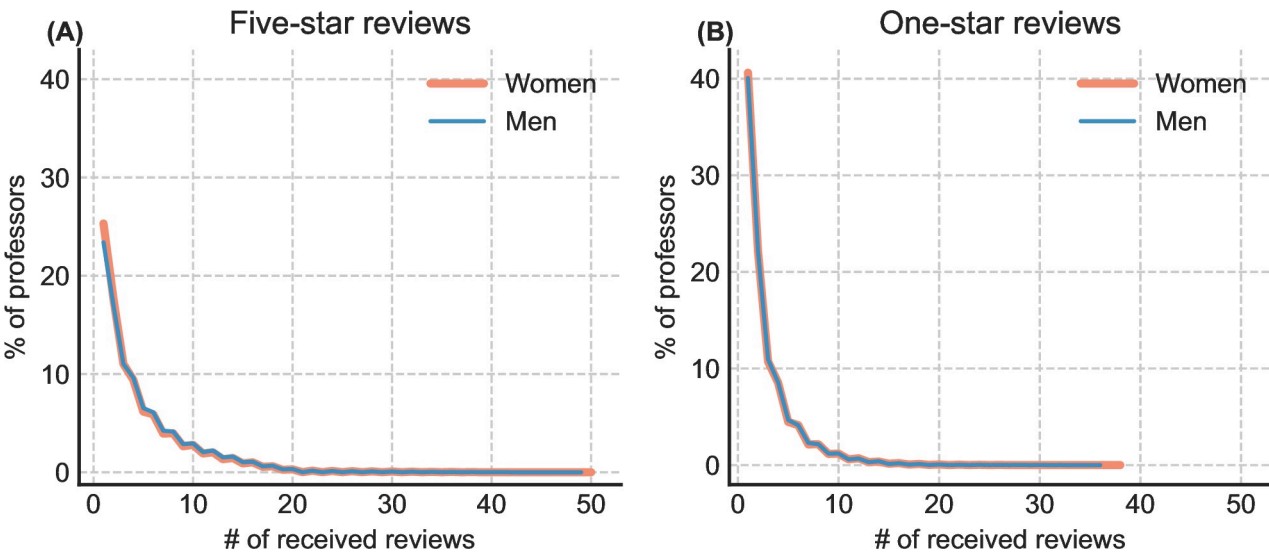

**Fig 3. Professors' percentage across the numbers of received reviews by professor gender and field.** (A) In five-star reviews. (B) In one-star reviews.

represent the most negative and positive evaluations on RMP, account for nearly half of the total reviews at 48.7%. Additionally, within the "good" (3.5–5 stars) and "bad" (1–2.4 stars) reviews as defined by RMP [56], five-star and one-star reviews make up 55.6% and 52.2%, respectively. This pattern was observed in all eight fields. Most significantly, women professors tend to receive a higher percentage of five-star reviews than men in most fields, except for *Medicine Health* and *Education*. On the other hand, women professors receive a higher percentage of one-star reviews than men in all fields, as shown in **Table 4**.

Another insight that emerged from our analysis is that, generally, the average ratings for women professors tend to be lower than those for men professors (see **Fig 4**). The multiple linear regression results show that ratings for women professors are significantly lower than those for men (coefficient < 0) in all fields except for Math & Computing, where no significant gender difference is detected (see **Table 5**). This finding provides evidence to support *H1*.

**Table 3. Average word counts of comments per review by field and gender.**

| Field | Five-star reviews | | | | One-star reviews | | | |
|---|---|---|---|---|---|---|---|---|
| | Women | Men | Difference | *p*-value* | Women | Men | Difference | *p*-value* |
| All | 37.45 | 36.48 | 0.97 | 0.000 | 39.84 | 39.04 | 0.80 | 0.000 |
| Applied Sciences | 36.38 | 35.43 | 0.95 | 0.000 | 39.47 | 38.84 | 0.63 | 0.000 |
| Natural Sciences | 41.10 | 39.04 | 2.06 | 0.000 | 42.63 | 41.26 | 1.37 | 0.000 |
| Math & Computing | 38.14 | 36.61 | 1.53 | 0.000 | 40.69 | 39.88 | 0.81 | 0.000 |
| Engineering | 31.70 | 30.73 | 0.97 | 0.000 | 35.12 | 34.34 | 0.78 | 0.035 |
| Medicine Health | 35.75 | 34.05 | 1.70 | 0.000 | 39.10 | 37.66 | 1.44 | 0.000 |
| Social Sciences | 37.88 | 36.82 | 1.06 | 0.000 | 40.11 | 38.98 | 1.14 | 0.000 |
| Education | 34.39 | 32.74 | 1.65 | 0.000 | 37.66 | 35.62 | 2.04 | 0.000 |
| Humanities | 37.16 | 36.69 | 0.47 | 0.000 | 39.08 | 38.43 | 0.65 | 0.000 |

*P-values were based on Mann-Whitney U tests.

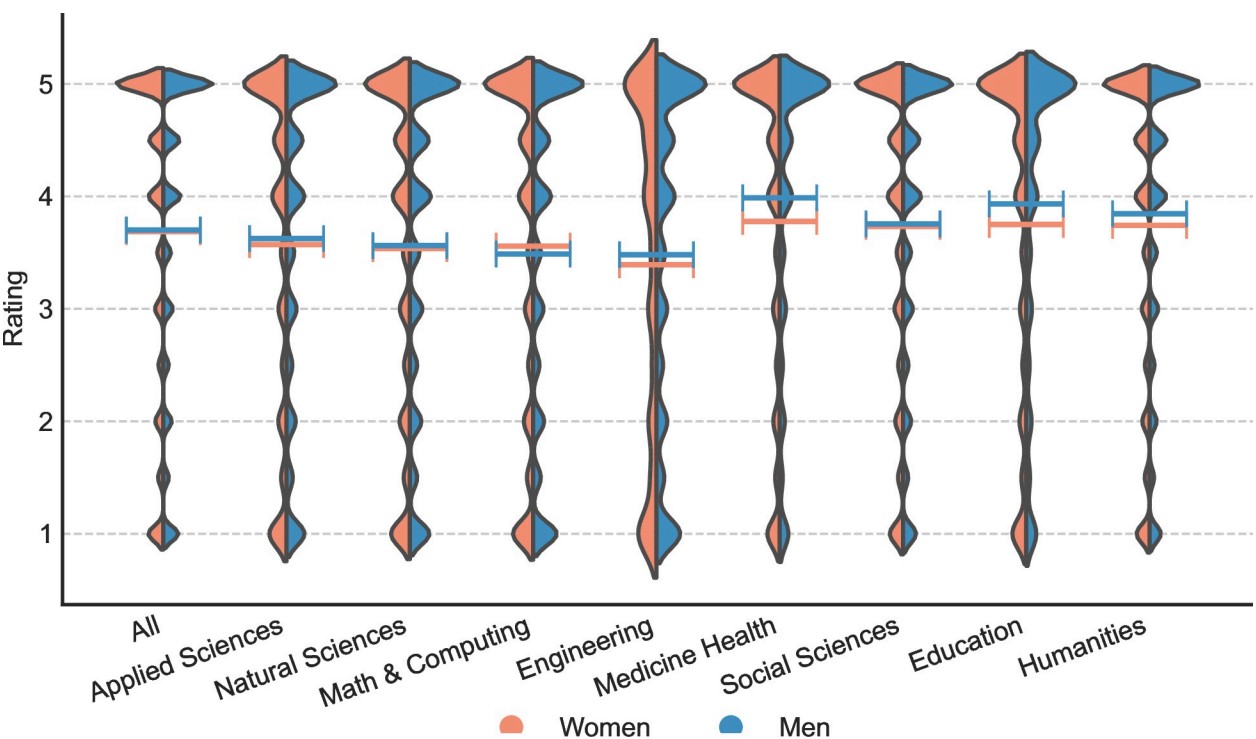

**Fig 4. Rating distributions and means of by professor gender and field.** Horizontal Lines in violins denote the mean values.

### Gender disparities in comment texts

**Differences in word usage.** Fig 5 presents the top 30 gender-distinct adjectives by Dunning score in RMP comments, along with the percentages of professors whose comments contain those adjectives. The percentages can help indicate the popularity of the adjective in the comments for professors. Our results show that students use adjectives that align with common gender role expectations in RMP reviews. Among five-star reviews, the most distinct and frequent adjectives for women professors tend to describe their kindness and supportiveness (*sweet*, *helpful*, *caring*, *willing*, and *kind*) and physical attractiveness (*lovely*, *pretty*, and *beautiful*). Women are also called out for being feminists. Conversely, in five-star reviews for men

**Table 4. Distributions of five- and one-star review numbers and percentages by professor gender and field.**

| Field | Five-star reviews | | | | One-star reviews | | | |
|---|---|---|---|---|---|---|---|---|
| | Women | | Men | | Women | | Men | |
| | # of reviews | Avg. % | # of reviews | Avg. % | # of reviews | Avg. % | # of reviews | Avg. % |
| Applied Sciences | 98,647 | 51.46 | 189,045 | 49.59 | 39,750 | 32.43 | 68,177 | 29.73 |
| Natural Sciences | 127,028 | 46.69 | 225,428 | 43.41 | 53,418 | 29.83 | 88,438 | 26.91 |
| Math & Computing | 147,998 | 50.03 | 259,265 | 46.30 | 61,325 | 31.00 | 121,722 | 30.37 |
| Engineering | 9,022 | 48.88 | 63,260 | 48.56 | 5,158 | 37.33 | 30,469 | 33.07 |
| Medicine Health | 96,579 | 60.38 | 61,289 | 61.87 | 25,901 | 36.21 | 10,659 | 31.26 |
| Social Sciences | 338,736 | 51.46 | 424,175 | 48.15 | 94,818 | 27.21 | 111,756 | 24.23 |
| Education | 68,092 | 59.96 | 38,788 | 60.45 | 19,614 | 35.24 | 7,846 | 31.79 |
| Humanities | 553,353 | 51.65 | 629,837 | 50.24 | 156,477 | 27.04 | 139,710 | 23.35 |
| Total | 1,439,455 | 52.12 | 1,891,087 | 48.89 | 456,461 | 29.39 | 578,777 | 26.82 |

**Table 5. Regression results for ratings by field.**

| Field | Coefficient (women-men) | p-value | 95% confidence interval |
|---|---|---|---|
| Applied Sciences | -0.077 | 0.000 | [-0.090, -0.063] |
| Education | -0.165 | 0.000 | [-0.190, -0.140] |
| Engineering | -0.132 | 0.000 | [-0.170, -0.094] |
| Humanities | -0.074 | 0.000 | [-0.081, -0.068] |
| Math & Computing | 0.004 | 0.463 | [-0.007, 0.016] |
| Medicine Health | -0.119 | 0.000 | [-0.138, -0.100] |
| Natural Sciences | -0.038 | 0.000 | [-0.050, -0.026] |
| Social Sciences | -0.046 | 0.000 | [-0.054, -0.038] |
| All | -0.052 | 0.000 | [-0.056, -0.048] |

professors, students tend to highlight their entertainment value (*funny*, *hilarious*, *interesting*, *entertaining*, *humorous*, and *amusing*), physical attractiveness (*handsome*), and intelligence (*smart*, *brilliant*, *knowledgeable*, and *intelligent*). Among one-star reviews, the most distinct adjectives for women are related to their lack of organizational skills (*not organized*, *not clear*, *scatterbrained*, and *not consistent*), unpleasant behavior or personality (*rude*, *mean*, *picky*, *strict*, *not helpful*, and *moody*), and lack of expertise (*elementary*, *not professional*, and *not clear*). On the other hand, negative reviews for men professors often pertain to the difficulty of the course (*hard*, *difficult*, *boring*, and *monotone*) or their arrogance (*arrogant*, *pompous*, and *cocky*).

Additionally, RMP ratings for men professors appear more generous than for women. Our results show that traditionally negative words (e.g., *dry*, *boring*, *arrogant*, and *corny*) also frequently appear in five-star reviews for men. Nevertheless, only positive words appear distinctively in five-star reviews for women professors. This suggests that students may rate men professors with five-star ratings, even when the corresponding review texts show signs of negativity.

The top 30 gender-distinct non-adjectives by Dunning score in RMP comments reflect a similar pattern (see **Fig 6**). We found that five-star reviews are more likely to use words related to supportiveness and emotion to describe women professors (*help*, *care*, *feedback*, *love*, and *sweetheart*) but use words related to entertainment (*joke*, *humor*, and *entertain*) and intelligence (*genius*, *doctor*, *theory*, and *knowledge*) to describe men. Furthermore, women professors are more likely to be called teachers and associated with teaching-related work (*work*, *assignment*, *project*, *group*, and *activity*). In contrast, men are more likely to be addressed as *professors*. Among one-star reviews, words related to families (*husband* and *child*) also distinctively appear for women professors. For men, signaling words for complaints are about lecturing (teach, *mumble*, and *ramble*) and tests (*test*, *curve*, *exam*, *midterm*, *quiz*, *book*, and *note*). This suggests the tendency of negative RMP reviews to focus more on the personal aspects of women professors than teaching for men.

**Differences in comment topics.** The results from the multiple linear regression analysis of the topic modeling show significant gender differences in the likelihood of professors receiving comments about certain topics and dimensions (see **Figs 7A and 8A**). For example, women professors are more likely than men to receive comments regarding the *Overall* dimension among five-star reviews. Specifically, men are more likely to be referred to as *professors*, while women are more likely to be referred to as *teachers* when provided an overall assessment among five-star reviews. Moreover, women are also more likely than men to receive comments highlighting their gender identity, as reflected by topics such as *woman; lady; teacher*. Under the *Teaching* dimension, women are more likely to receive comments related

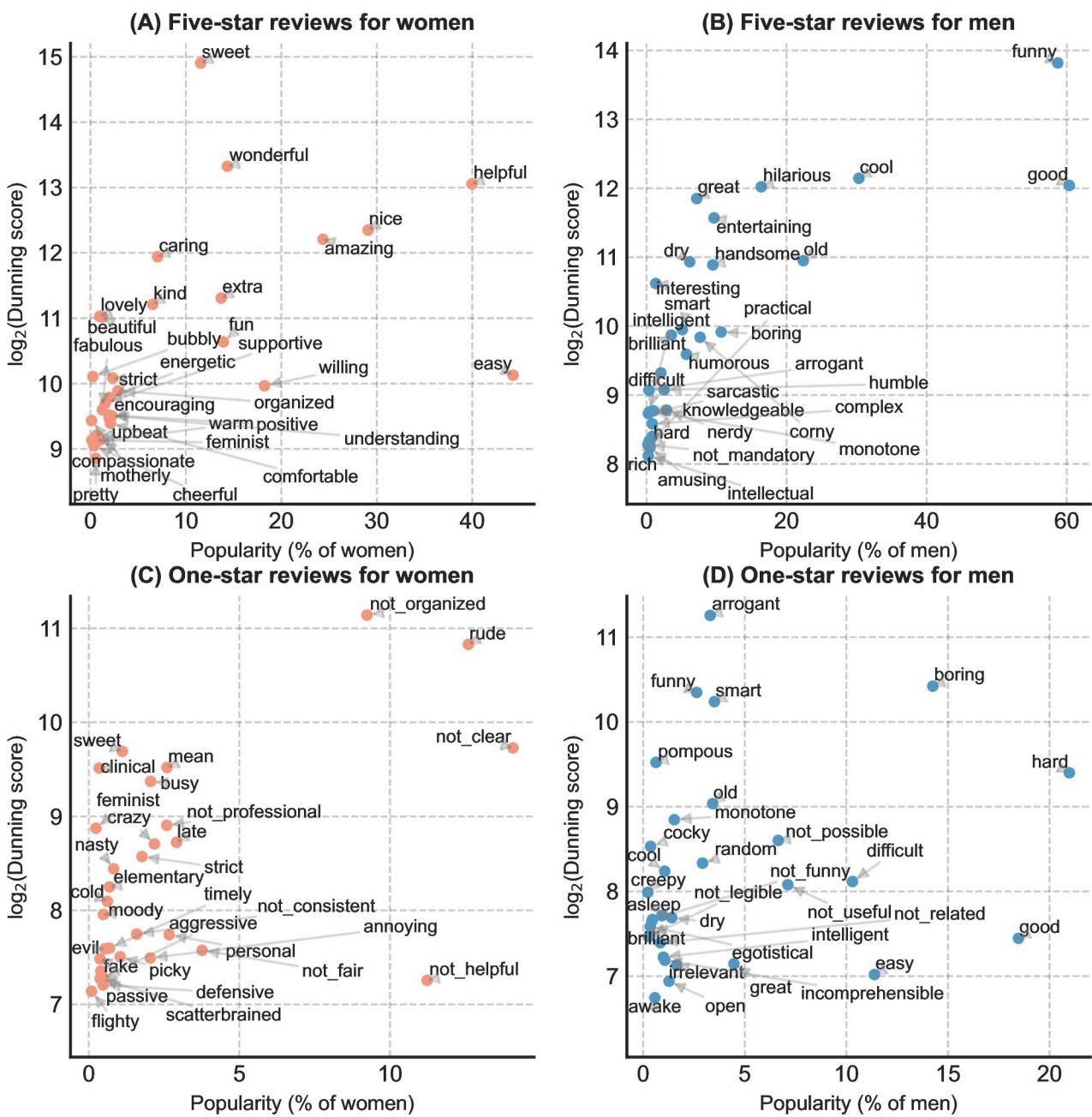

**Fig 5. Dunning score and popularity of the top 30 gender-distinct adjectives in RMP comments by gender and rating.** The popularity of a word is the percentage of professors whose comments contain the word. (A) In five-star reviews for women. (B) In five-star reviews for men. (C) In one-star reviews for women. (D) In one-star reviews for men.

to helping, caring, and support, as reflected by explaining materials, feedback, treats, and responding to emails. On the other hand, comments for men are more likely to praise their lecturing-related performance, as shown by topics about lectures and storytelling. Among comments concerning the *Personal* dimension, men are more likely than women to be praised for their entertainment value, intelligence, accent, and physical attractiveness. In topics related to course materials (*Material*), syllabus-related issues are more likely to be mentioned in comments for women, with textbook-related topics more likely for men. Women are also more

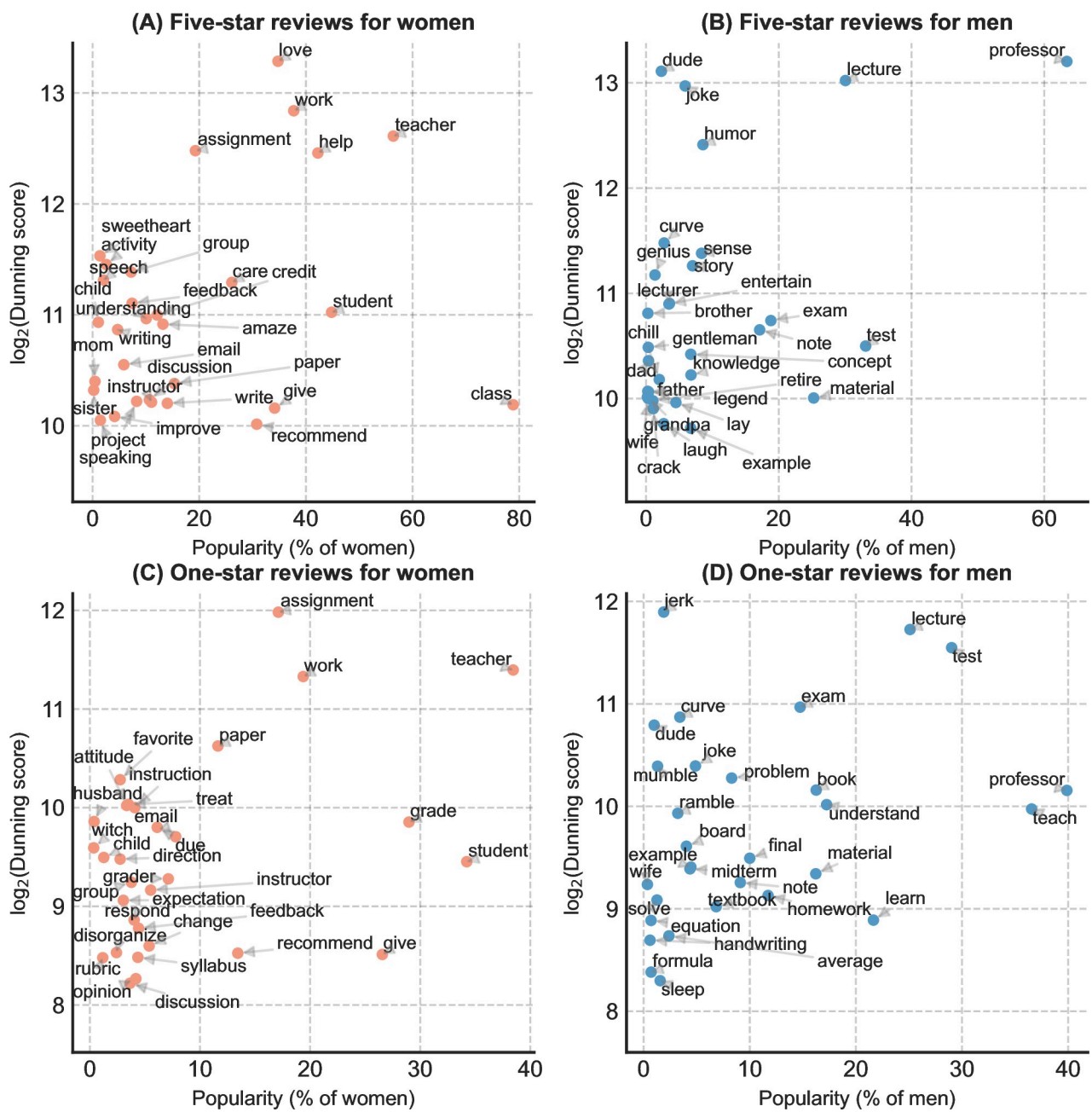

**Fig 6. Dunning score and popularity of the top 30 gender-distinct non-adjectives in RMP comments by gender and rating.** The popularity of a word is the percentage of professors whose comments contain the word. (A) In five-star reviews for women. (B) In five-star reviews for men. (C) In one-star reviews for women. (D) In one-star reviews for men.

likely to be commented on course structure-related topics (*Structure*), including workload and study effort.

In one-star reviews, the pattern of gender differences in comments is similar to that of five-star reviews, with a few exceptions (see **Figs 7B and 8B**). A key difference exists in the *Personal* dimension, where the topics for women professors tend to contain more negative words and phrases. Women are more likely to be described as rude and playing favorites in these comments. Additionally, a specific topic under the *Personal* dimension associates women

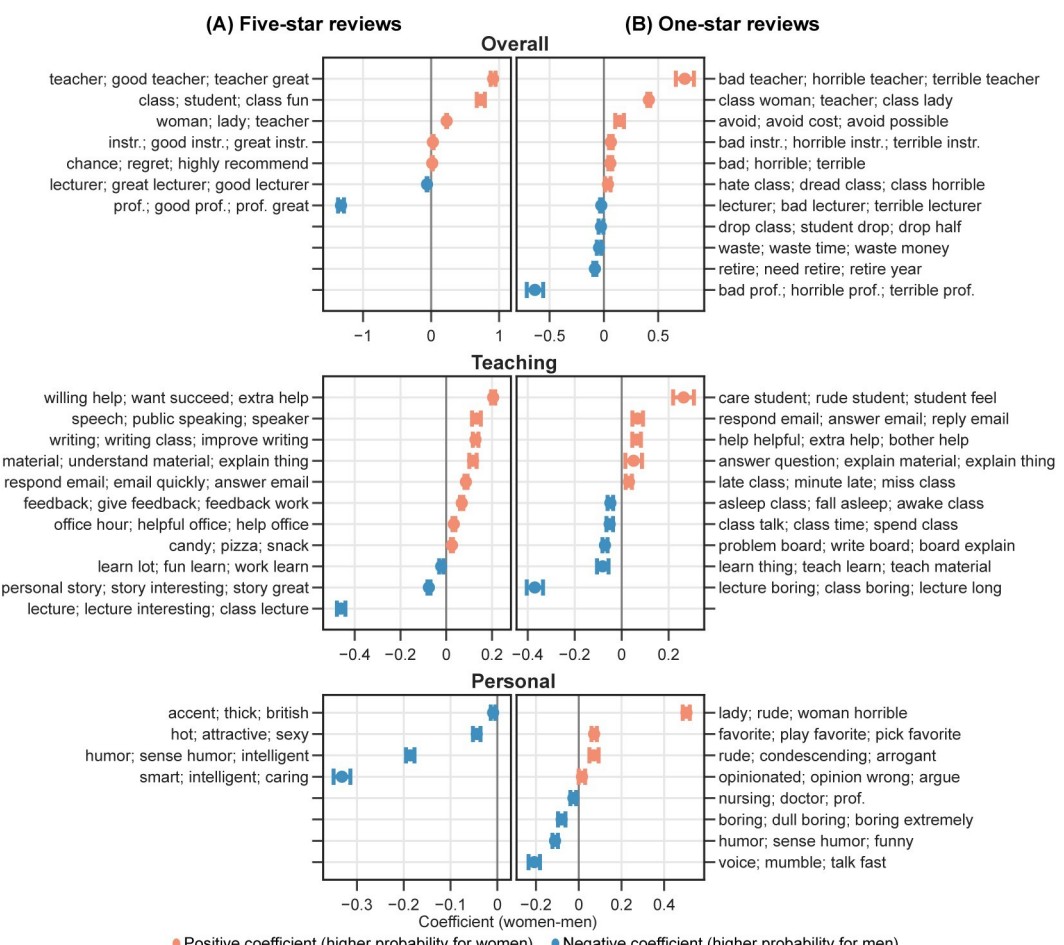

**Fig 7.** Regression analysis of predicted possibilities from topic modeling in (A) five-star and (B) one-star reviews by professor-level topic dimensions. An orange mark denotes a positive coefficient of gender (1 = women), meaning that the topic is more likely to appear in comments for women professors. A blue mark denotes a negative coefficient of gender, indicating that the topic is more likely to appear in comments for men professors. Only statistically significant results are shown. prof. = professor; instr. = instructor.

professors more frequently with being "rude" and "horrible." In contrast, under the same dimension, we did not see topics from one-star reviews on men professors containing extremely critical words and phrases.

We also performed topic analysis on comments at the field level. Overall, the above findings hold across all fields, with only a limited number of topics varying by field. We visualized topic analysis results by field in an interactive figure for further investigation (https://mybinder.org/v2/gh/x-zhe/RMP_gendered_comment/HEAD?urlpath = %2Fvoila%2Frender%2FRMPreg.ipynb). Overall, the results from the word usage analysis and topic analysis support *H2*.

**Sentiment analysis of topics in comments.** Consistent with *H4*, our sentiment analysis suggests that gender plays a significant role in the RMP comments' language sentiments. Among the 43 topics (excluding *Subject* and *Other/not specified*) identified from five-star comments, the levels of positivity for 37 topics are significantly associated with professors' genders (see **Figs 9A and 10A**). The levels of negativity for 46 (out of 67) topics from one-star reviews are significantly associated with professors' genders. We refer to these 37 and 46 topics as gender-significant topics (see **Figs 9B and 10B**).

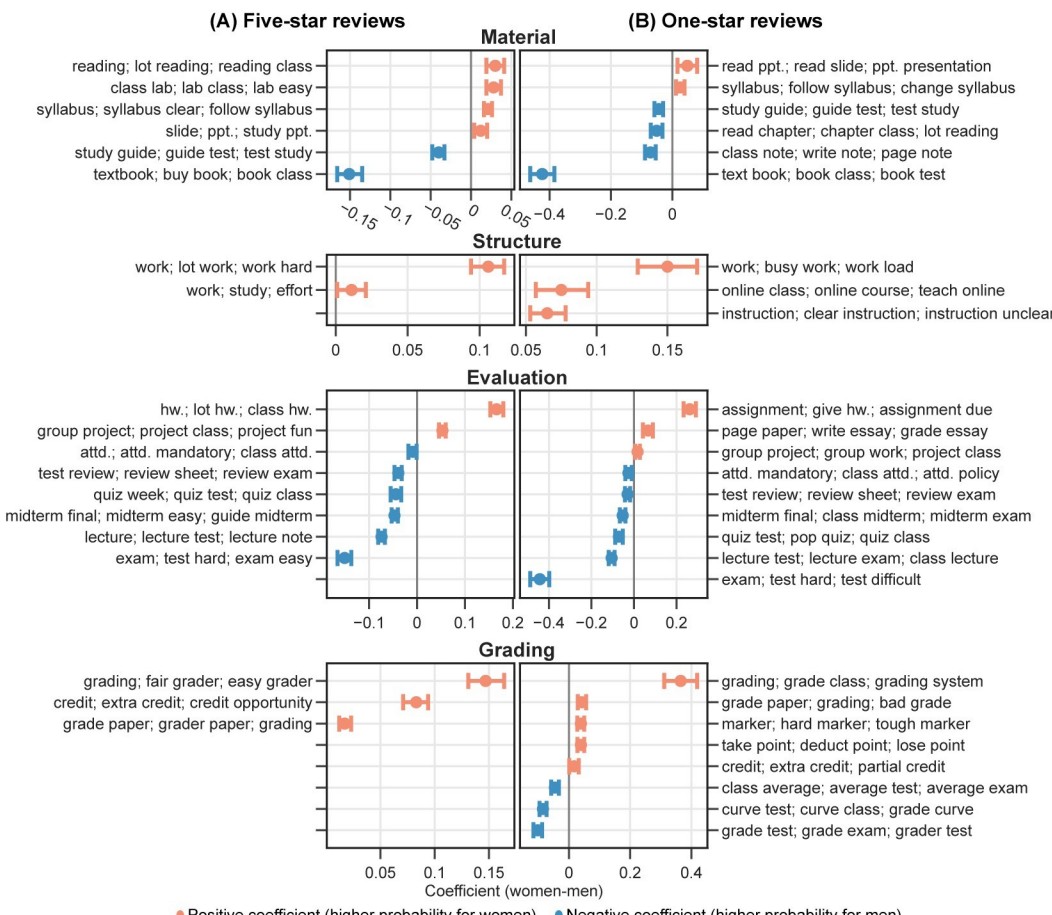

**Fig 8.** Regression analysis of predicted possibilities from topic modeling in (A) five-star and (B) one-star reviews by course-level topic dimensions. *Subject* dimension is excluded. Figure encodings have the same meanings as Fig 7. Only statistically significant results are shown. ppt. = powerpoint; hw. = homework; attd. = attendance.

Furthermore, we showed that comments for women professors are of higher sentimental polarity in both five- and one-star reviews than those for men. Specifically, five-star comments for women professors are more positive than those for men among gender-significant topics, as suggested by the significantly higher sentiment scores on these topics for women. The only exception is the topic labeled "*lecture; great/good lecturer,*" which is significantly more positive for men. In one-star reviews, the negative comments for women are more negative than those for men among almost all gender-significant topics under the nine dimensions, as suggested by the significantly higher negative sentiment scores on these topics for women. One exception is the topic concerning playing favorite under the *Personal* dimension, which appears to be more negative for men. Likewise, we performed the sentiment analysis by field and found consistent results. These results are also available in the aforementioned interactive figure.

## Discussion

Our analysis of RMP ratings and comments found evidence of gender disparities in SET. We found that women professors consistently receive lower ratings than men on RMP across most fields, which confirms previous studies that have shown similar results in multiple fields and institutions [17,24,25,78,79]. The word usage and topic analysis of the comments provide a

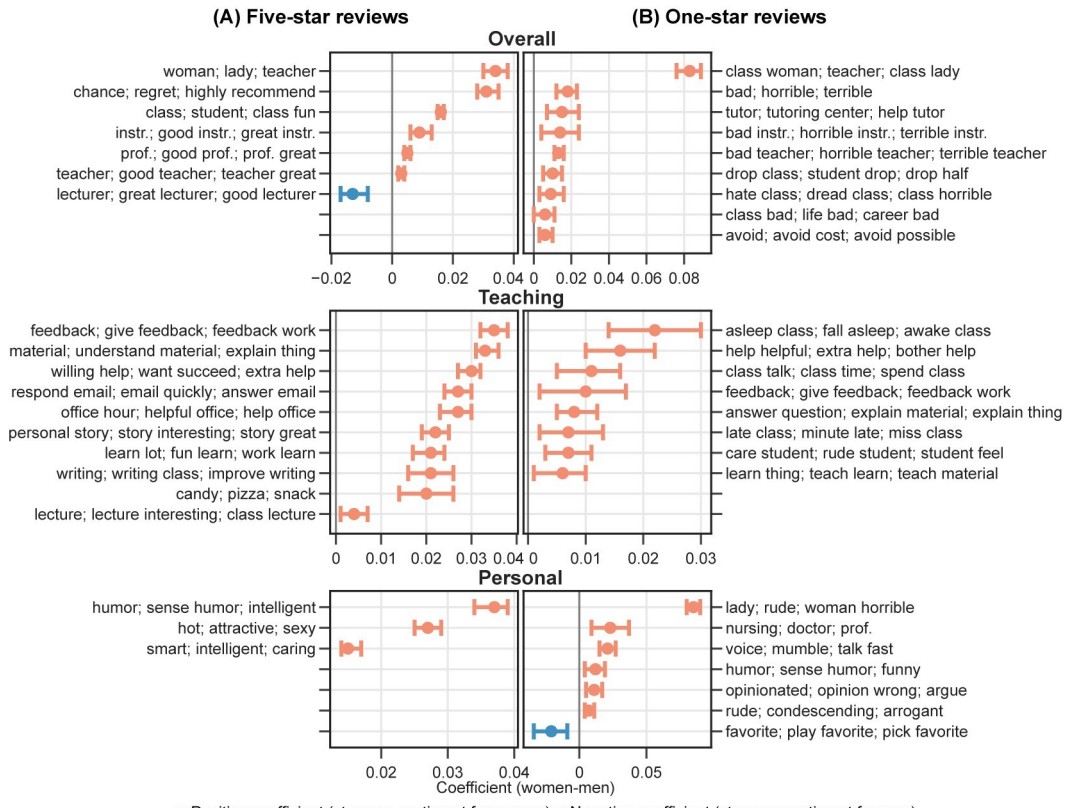

**Fig 9. Regression analysis of predicted sentiment scores by professor-level topic dimensions.** Orange and blue marks denote positive and negative coefficients of gender (1 = women), respectively. Only gender-significant topics where gender is a significant contributing factor in the regression analysis are shown. prof. = professor; instr. = instructor. (A) Regression analysis of five-star reviews based on positive sentiment scores. (B) Regression analysis of one-star reviews based on negative sentiment scores.

comprehensive view of gendered teaching evaluations for professors in higher education. Consistent with previous findings [80], the word usage analysis shows that women are often described with adjectives related to their physical attractiveness, supportiveness, and organizational skills. In contrast, men are described with adjectives about entertainment value, intelligence, course difficulty, physical attractiveness, and arrogance. The non-adjectives additionally reveal that women are more likely to be associated with teaching work and be criticized for their leadership, assignments, personal attitude, and family issues. At the same time, men are more likely to receive complaints about tests and grading.

Using topic modeling techniques, we found that for the *Overall* dimension, women professors are more likely to be addressed as a teacher or an instructor rather than a professor or a lecturer, which is consistent with previous findings [12]. The presence of woman-related topics and absence of man-related topics in the *Overall* dimension for both five- and one-star reviews also implies that students may pay more attention to women professors' gender in comments. In the *Teaching* dimension, students focus more on women's supportiveness and responsiveness and men's lecture delivery effects. The findings in the *Personal* dimension indicate that women are more likely to be criticized for their uncaring or unfair attitude to students, and men are more likely to receive comments about their intelligence and humor. Notably, although both women and men are associated with gender-specific adjectives describing their physical attractiveness (e.g., *beautiful* for women and *handsome* for men), men are more likely

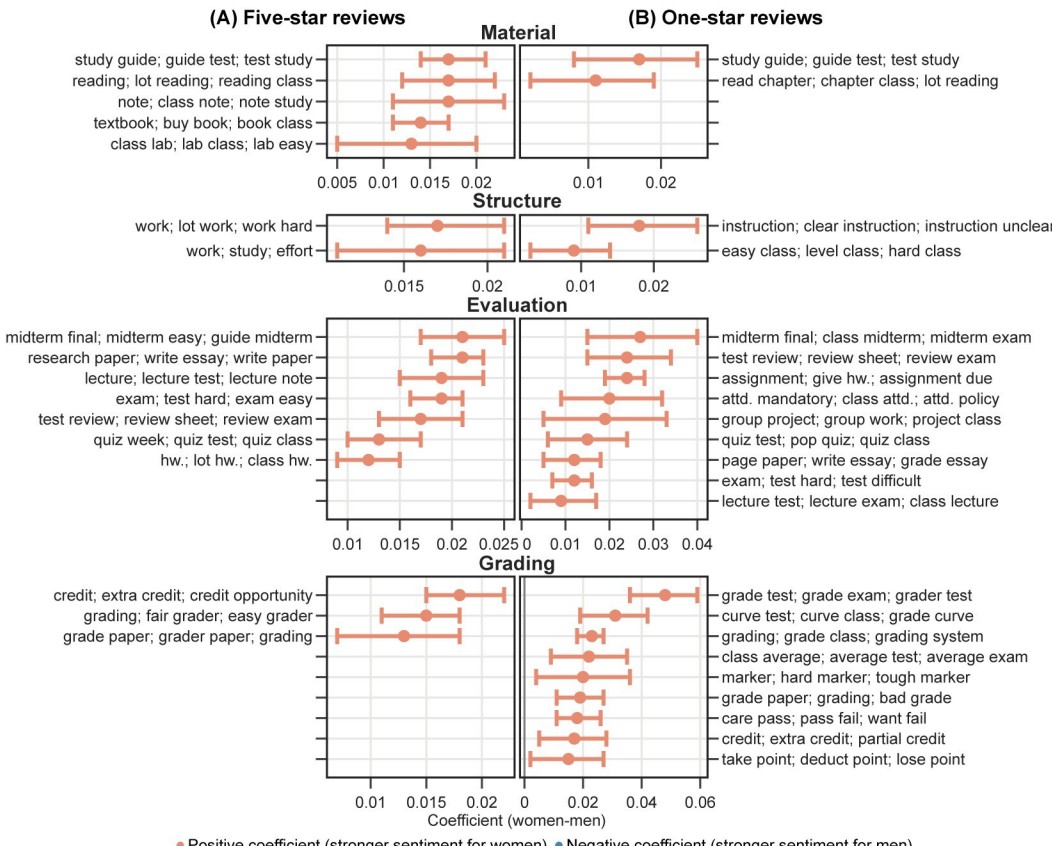

**Fig 10. Regression analysis of predicted sentiment scores by course-level topic dimensions.** Figure encodings have the same meanings as Fig 9. Only gender-significant topics are shown. *Subject* dimension is excluded. hw. = homework; attd. = attendance. (A) Regression analysis of five-star reviews based on positive sentiment scores. (B) Regression analysis of one-star reviews based on negative sentiment scores.

to be commented on for their physical attractiveness by general description, as shown by the topic *hot; attractive; sexy*. It is likely that the perception of a professor's attractiveness is correlated with the perception of their expertise [60,81]. Associated more with "smart," men professors may tend to establish a "smart and sexy" image to students and boost their attractiveness.

The other dimensions also exhibit consistent gender disparities in comments. Overall, women professors are more likely to be commented on for assignments, reading, papers/ essays, and group projects, while men are for test-related issues. Under the *Structure*, *Evaluation*, and *Grading* dimensions, women professors receive more comments about workload and grading practice. It suggests that students more often criticize women professors for workload and men for difficulty levels. Although not directly implied, we consider this to be related to students' expectations based on the gender of professors. Students may assume that, as "caregivers" and "supporters," women professors should relieve students of heavy workloads and give them "easy" high grades [16,53]. It is also likely that women are more commented about assignments for not organizing them well, giving clear instructions, or following the syllabus, as reflected in the topic modeling results.

Furthermore, multiple analyses in our study suggest that women receive more polarized ratings and reviews than men. Women professors have a higher percentage of both five-star and one-star reviews across most fields. The word usage analysis shows that five-star reviews for men professors often mention negative words like *dry* and *boring*, and positive words such

as *good* and *funny* also appear as frequent gender-specific words in one-star reviews for men. Most importantly, sentiment analysis reveals that for most comment topics, compared with comments for men, comments for women professors are more positive in five-star reviews and more negative in one-star reviews. This aligns with a previous finding that reviews for men TED presenters are more neutral, while those for women are more polarized [82]. Consistent with the shifting standards theory, the polarity of women's received reviews indicates that students' subjective evaluations and standards are influenced by the professor's gender, not a set standard. However, the polarized opinions for women do not depend on the comment topic's relation to gender roles and stereotypes. We assume that the stereotype driving this subjectivity of judgments is from the general linkage between women and "emotional" [13], which reduces comments' neutrality.

## Theoretical and practical implications

This study provides both theoretical and practical implications for understanding gender disparities in SET and higher education broadly. We proposed a novel pipeline to contextualize the gender disparities in SET, including a new approach for gender imputation based on gender pronounces and titles and a combination of rating analysis and comment analysis using advanced natural language processing techniques. In addition, extending from the role congruity theory, we systematically identified and quantified the gendered patterns of word usage and topics extracted from comments, which exhibits a highly fine-grained scope into the gender role expectations held by students toward professors. Many gender-specific words and topics are worth further analysis to uncover underlying gender role expectations. Moreover, we contribute to the application of shifting standards theory to gender disparities in SET by showing that students' evaluations of women professors are more polarized than men, regardless of the comment topics. This suggests that the ratings and language used in SET evaluations for different genders do not conform to the same standard and may be influenced by systemic stereotypes of women, independent of the context of the review.

Our findings of gender disparities embedded in both ratings and comments in SET highlight the need for caution in using SET for assessments and decision-making concerning professors. Overreliance on SET results may reinforce gender role expectations and put pressure on professors to conform to these roles [19]. Our results also aim to raise awareness among students of the potential for explicit and implicit gender biases in their evaluations of professors, as sometimes students do not realize the biases until specifically informed [14]. We encourage students to consider the potential for such biases in RMP reviews when browsing professors' reviews and making course enrollment decisions accordingly.

## Limitations and future work suggestions

Our study sheds light on gender disparities in RMP reviews for professors but is inevitably limited by the data source and methods. As mentioned by He et al. [19], our dataset of RMP reviews may be prone to self-selection biases, which potentially affect the validity of the results. The results may also be inapplicable to the SET within higher education systems different from North America. Additionally, the gender imputation algorithm used did not allow us to identify non-binary genders, and the anonymity of evaluators on RMP prevented us from considering the influence of students' gender on their perceptions of professors [7,16,54]. This study only analyzed English comments, missing information from non-English evaluations. Limited by data availability, the regression models used in the analysis may omit key variables that affect the dependent variable. Future work could improve the accuracy of results by using more comprehensive and informative SET datasets, considering non-binary genders, and

extending the analysis to multilingual SET reviews. Further exploration of gender stereotypes and sentiment polarity revealed in this study is also needed.

## Supporting information

**S1 Table. Top 10 departments in each field.**
(DOCX)

**S2 Table. Topics in five- and one-star reviews and statistics of their probabilities.** Representative sentences have been replaced with gender-neutral pronouns. (A) five-star reviews. (B) one-star reviews.
(DOCX)

## Acknowledgments

We appreciate the valuable comments and suggestions on the manuscript from anonymous reviewers.

## Author Contributions

**Conceptualization:** Xiang Zheng, Shreyas Vastrad, Chaoqun Ni.

**Data curation:** Xiang Zheng, Shreyas Vastrad, Jibo He, Chaoqun Ni.

**Formal analysis:** Xiang Zheng, Shreyas Vastrad, Chaoqun Ni.

**Investigation:** Xiang Zheng, Shreyas Vastrad, Chaoqun Ni.

**Methodology:** Xiang Zheng, Shreyas Vastrad, Chaoqun Ni.

**Project administration:** Chaoqun Ni.

**Resources:** Jibo He, Chaoqun Ni.

**Software:** Xiang Zheng, Shreyas Vastrad.

**Supervision:** Chaoqun Ni.

**Validation:** Xiang Zheng, Jibo He, Chaoqun Ni.

**Visualization:** Xiang Zheng, Jibo He, Chaoqun Ni.

**Writing – original draft:** Xiang Zheng, Chaoqun Ni.

**Writing – review & editing:** Xiang Zheng, Shreyas Vastrad, Jibo He, Chaoqun Ni.

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
