## [Decision Letter · Decision Letter 0]

24 Jan 2023

PONE-D-22-35274Contextualizing gender disparities in online teaching evaluations for professorsPLOS ONE

Dear Dr. Chaoqun Ni

Thank you for submitting your manuscript to PLOS ONE. After careful consideration, we feel that it has merit but does not fully meet PLOS ONE’s publication criteria as it currently stands. Therefore, we invite you to submit a revised version of the manuscript that addresses the points raised during the review process.

ACADEMIC EDITOR:After reading the article and the response given by the two reviewers, I agree with them that the article meets the characteristics that make it publishable in PLOS One, however, it needs to undergo some improvements. 

Therefore, I request the authors to carry out the improvements proposed by the two reviewers.

We look forward to receiving your revised manuscript.

Kind regards,

José Manuel Santos Jaén

Academic Editor

PLOS ONE

3. Please remove your figures from within your manuscript file, leaving only the individual TIFF/EPS image files, uploaded separately. These will be automatically included in the reviewers’ PDF.

Additional Editor Comments:

After reading the article and the response given by the two reviewers, I agree with them that the article meets the characteristics that make it publishable in PLOS One, however, it needs to undergo some improvements.

Therefore, I request the authors to carry out the improvements proposed by the two reviewers.

Reviewers' comments:

Reviewer's Responses to Questions

**Comments to the Author**

1. Is the manuscript technically sound, and do the data support the conclusions?

Reviewer #1: Partly

Reviewer #2: Partly

2. Has the statistical analysis been performed appropriately and rigorously? 

Reviewer #1: No

Reviewer #2: N/A

3. Have the authors made all data underlying the findings in their manuscript fully available?

Reviewer #1: Yes

Reviewer #2: Yes

4. Is the manuscript presented in an intelligible fashion and written in standard English?

Reviewer #1: No

Reviewer #2: Yes

5. Review Comments to the Author

Reviewer #1: Thank you for the opportunity to review this manuscript. This paper demonstrates the attempt to showcase the issue of gender disparity in relation to student evaluation of teaching, or SET. However, in order to enhance the scholarly rigour of the manuscript, the authors could consider refining it based on these suggestions:

1. The educational issue, problem formulation, and problem statement, as well as the study's contextualisation, were not discussed in depth in the introduction section. There is a lack of recent literature that discusses the gender disparity in academia, particularly within the context of this study.

2. The knowledge/research gap, particularly the study of recent research trends within the research context, has to be explored more critically.

3. Since the research is quantitative, the structure of the research could be changed by adding the research objectives, research questions, and hypotheses that reflect the problem formulation and problem description. This is followed by the methodology and analysis sections, and ends with the presentations of the findings.

4. Theoretical and conceptual frameworks need to be added and thoroughly discussed.

5. The methodology part needs to be discussed in-depth. This includes the research design, the research instrumentation, and the validation of the data.

6. Ethical considerations must be thoroughly discussed, and especially in relation to the data protection act of the countries from which the data was derived.

Reviewer #2: The paper “Contextualizing gender disparities in online teaching evaluations for professors” is within scope of the Plos One journal. I found several shortcomings in this work and have suggestion to the authors to consider it i.e.,

1. Abstract section: Authors are requested to add detail tools and techniques used data analysis and production of the analyzed results.

2. Restructure heading of this paper i.e., (1) Introduction, (2) Methods, (3) Results, (4) Discussion and conclusion, (5) Theoretical contribution and practical implication, and (6) Limitations and future work suggestion

3. After “Discussion and conclusion” section, authors establish new section i.e., “Theoretical contribution and practical implication” and thoroughly explain.

4. Furthermore, last heading must be “Limitations and future work suggestion”. In this section, authors explain possible limitation of current study and how author researchers can extend this work.

5. To further improve contents of this paper, authors are requested to review and cite below mentioned paper (if possible):

Abbas, A., Haruna, H., Arrona-Palacios, A., Camacho-Zuñiga, C., Núñez-Daruich, S., Enríquez de la O, J. F., ... & Hosseini, S. (2022). Students’ evaluations of teachers and recommendation based on course structure or teaching approaches: An empirical study based on the institutional dataset of student opinion survey. Education and Information Technologies, 1-16.

Abbas, A., Arrona-Palacios, A., Haruna, H., & Alvarez-Sosa, D. (2020). Elements of students’ expectation towards teacher-student research collaboration in higher education. In 2020 IEEE Frontiers in Education Conference (FIE) (pp. 1-5). IEEE.

6. PLOS authors have the option to publish the peer review history of their article (what does this mean?). If published, this will include your full peer review and any attached files.

Reviewer #1: No

Reviewer #2: No

---

## [Author Response · Author response to Decision Letter 0]

13 Feb 2023

Response to Reviewers

RE:

PONE-D-22-35274

Contextualizing gender disparities in online teaching evaluations for professors

PLOS ONE

Dear Dr. Chaoqun Ni

Thank you for submitting your manuscript to PLOS ONE. After careful consideration, we feel that it has merit but does not fully meet PLOS ONE’s publication criteria as it currently stands. Therefore, we invite you to submit a revised version of the manuscript that addresses the points raised during the review process.

ACADEMIC EDITOR:

After reading the article and the response given by the two reviewers, I agree with them that the article meets the characteristics that make it publishable in PLOS One, however, it needs to undergo some improvements. 

Therefore, I request the authors to carry out the improvements proposed by the two reviewers.

We highly appreciate the comments and suggestions from the editors and reviewers. We have taken every step to address and respond to the reviewers’ comments, which we feel has improved our manuscript. Below, the reviewer and editor’s comments are in black, whereas our responses are in blue. We have also included a version of our manuscript with differences highlighted (though without references and figure labels). Moreover, we made minor changes throughout in order to improve the quality of the manuscript. 

We first respond to each reviewer’s comments and then respond to editorial comments at the end. 

 

Reviewers’ comments:

Reviewer #1: 

1. The educational issue, problem formulation, and problem statement, as well as the study’s contextualisation, were not discussed in depth in the introduction section. There is a lack of recent literature that discusses the gender disparity in academia, particularly within the context of this study.

We thank the reviewer for the comments. To address your concerns, we have made the following changes: 

1) We extended our problem statements in the first paragraph of the “Introduction” Section. 

2) We further elaborated on the problems, the background, and the contexts in the newly added “Related research and theoretical framework” section.

3) We discussed the gender disparity issue in academia in the context of this particular study in the “Gender disparities in academia and teaching evaluation” subsection under the “Related research and theoretical framework” section.

2. The knowledge/research gap, particularly the study of recent research trends within the research context, has to be explored more critically.

We thank the reviewer for the suggestions. We added a new section, “Related research and theoretical framework,” to the manuscript to review the recent trends in RateMyProfessors-related research and gender biases in SET. Following the suggestion, we summarized and discussed the current research gap accordingly.

3. Since the research is quantitative, the structure of the research could be changed by adding the research objectives, research questions, and hypotheses that reflect the problem formulation and problem description. This is followed by the methodology and analysis sections, and ends with the presentations of the findings.

We thank the reviewer for the thoughtful suggestions. We have highlighted our research objectives and questions in the last paragraph of the “Introduction” section. We also formulate our hypotheses in the “Related research and theoretical framework” section after introducing current studies and theoretical framework.

4. Theoretical and conceptual frameworks need to be added and thoroughly discussed.

We thank the reviewer for the thoughtful comment. The newly appended “Related research and theoretical framework” section now discusses the theoretical frameworks that help formulate our research design and interpret the results.

5. The methodology part needs to be discussed in-depth. This includes the research design, the research instrumentation, and the validation of the data.

We highly appreciate the suggestion. We added a subsection, “Data and procedures,” under the “Methods” section to introduce the grand research design of this study. We also added a “Data validation and sample representativeness” subsection to discuss the potential skewness and representativeness in the sample, as well as its impact on the validity of our results. For the purpose of easier reading, details of research instrumentation are now aggregated under the subsection “Empirical analysis” in the “Methods” section.

6. Ethical considerations must be thoroughly discussed, and especially in relation to the data protection act of the countries from which the data was derived.

We appreciate the reviewer’s comments regarding the ethical issues of the study. We always consider ethical conduct and privacy as one of our top priorities in the responsible conduct of research. We used a dataset that is publicly available in its nature. Furthermore, we also take the following steps to avoid exposing individuals in our analysis: (1) We focused on professors associated with institutions located in the United States; (2)We de-identified individual professors for all analyses in the manuscript after gender imputation; and (3) We analyzed the data at the level of gender and discipline groups instead of individuals, imposing minimum risk to individual human subjects. Per the guidelines provided by the International Review Board at the University of Wisconsin-Madison, our study is eligible for exemption from institutional review board reviews (45 CFR § 46.104 (d) (1-8)). Moreover, we also observed that PLOS ONE published a few studies (see the list below) using data from RateMyProfessors.com. We made sure our study followed similar approaches regarding data ethics.

The following discussion about the ethical considerations has been added to the “Data and procedures” subsection under “Methods.”

With data privacy and ethics in mind, we de-identified individual professors for all analyses in the manuscript after gender imputation, which relied on professors’ first names. Moreover, we analyzed the data at the level of gender categories instead of individuals, imposing minimum risk to human subjects. Additionally, RMP is publicly available. Per the guidelines provided by the International Review Board at the University of Wisconsin-Madison, this study is eligible for exemption from institutional review board reviews (45 CFR § 46.104 (d) (1-8)).

Other studies in PLOS ONE using RMP data: 

Murray D, Boothby C, Zhao H, Minik V, Bérubé N, Larivière V, et al. Exploring the personal and professional factors associated with student evaluations of tenure-track faculty. PLoS ONE. 2020;15: e0233515. doi:10.1371/journal.pone.0233515

Storage D, Horne Z, Cimpian A, Leslie S. The Frequency of “Brilliant” and “Genius” in Teaching Evaluations Predicts the Representation of Women and African Americans across Fields. PLOS ONE. 2016;11. doi:10.1371/journal.pone.0150194

Reviewer #2: 

1. Abstract section: Authors are requested to add detail tools and techniques used data analysis and production of the analyzed results.

We thank the reviewer for the thoughtful suggestions. We have added descriptions of related NLP techniques and statistical methods to the abstract. The current abstract reads as follows (with minor grammatical changes): 

Student evaluation of teaching (SET) is widely used to assess teaching effectiveness in higher education and can significantly influence professors’ career outcomes. Although earlier evidence suggests SET may suffer from biases due to the gender of professors, there is a lack of large-scale examination to understand how and why gender disparities occur in SET. This study aims to address this gap in SET by analyzing approximately 9 million SET reviews from RateMyProfessors.com under the theoretical frameworks of role congruity theory and shifting standards theory. Our multiple linear regression analysis of the SET numerical ratings confirms that women professors are generally rated lower than men in many fields. Using the Dunning log-likelihood test, we show that words used in student comments vary by the gender of professors. We then use BERTopic to extract the most frequent topics from one- and five-star reviews. Our regression analysis based on the topics reveals that the probabilities of specific topics appearing in SET comments are significantly associated with professors’ genders, which aligns with gender role expectations. Furtherly, sentiment analysis indicates that women professors’ comments are more positively or negatively polarized than men’s across most extracted topics, suggesting students’ evaluative standards are subject to professors’ gender. These findings contextualize the gender gap in SET ratings and caution the usage of SET in related decision-making to avoid potential systematic biases towards women professors.

2. Restructure heading of this paper i.e., (1) Introduction, (2) Methods, (3) Results, (4) Discussion and conclusion, (5) Theoretical contribution and practical implication, and (6) Limitations and future work suggestion

We thank the reviewer for the suggestion. We have restructured our manuscript into the following sections: Introduction, Related research and theoretical framework, Methods, Results, Discussion, Theoretical and practical implications, and Limitations and future work suggestions.

3. After “Discussion and conclusion” section, authors establish new section i.e., “Theoretical contribution and practical implication” and thoroughly explain.

We thank the reviewer for the suggestion. As mentioned in our previous response, a “Theoretical and practical implications” section has been added to the manuscript.

4. Furthermore, last heading must be “Limitations and future work suggestion”. In this section, authors explain possible limitation of current study and how author researchers can extend this work.

We thank the reviewer for the suggestion. As mentioned in our previous response, a “Limitations and future work suggestions” section has been added to the manuscript.

5. To further improve contents of this paper, authors are requested to review and cite below mentioned paper (if possible):

Abbas, A., Haruna, H., Arrona-Palacios, A., Camacho-Zuñiga, C., Núñez-Daruich, S., Enríquez de la O, J. F., ... & Hosseini, S. (2022). Students’ evaluations of teachers and recommendation based on course structure or teaching approaches: An empirical study based on the institutional dataset of student opinion survey. Education and Information Technologies, 1-16.

Abbas, A., Arrona-Palacios, A., Haruna, H., & Alvarez-Sosa, D. (2020). Elements of students’ expectation towards teacher-student research collaboration in higher education. In 2020 IEEE Frontiers in Education Conference (FIE) (pp. 1-5). IEEE.

We thank the reviewer for the recommendation of related research. We have carefully reviewed the papers and have added them to our “Introduction” and “Related research and theoretical framework” sections. 

Editorial Comments

1. Please ensure that your manuscript meets PLOS ONE’s style requirements, including those for file naming.

The manuscript has been formatted according to PLOS ONE’s style requirements.

2. In your Data Availability statement, you have not specified where the minimal data set underlying the results described in your manuscript can be found.

Minimal underlying de-identified datasets, web scraping scripts, and analytical codes are now available on GitHub (https://github.com/UWMadisonMetaScience/rmp).

3. Please remove your figures from within your manuscript file, leaving only the individual TIFF/EPS image files uploaded separately. These will be automatically included in the reviewers’ PDF.

Figures have been removed from the manuscript.

Captions for Supporting Information files are now included in the manuscript.

---

## [Decision Letter · Decision Letter 1]

20 Feb 2023

Contextualizing gender disparities in online teaching evaluations for professors

PONE-D-22-35274R1

Dear Dr. Ni,

We’re pleased to inform you that your manuscript has been judged scientifically suitable for publication and will be formally accepted for publication once it meets all outstanding technical requirements.

Kind regards,

José Manuel Santos Jaén

Academic Editor

PLOS ONE

Additional Editor Comments (optional):

Based on the review, I consider that the article can be published.

Reviewers' comments:

Reviewer's Responses to Questions

**Comments to the Author**

1. If the authors have adequately addressed your comments raised in a previous round of review and you feel that this manuscript is now acceptable for publication, you may indicate that here to bypass the “Comments to the Author” section, enter your conflict of interest statement in the “Confidential to Editor” section, and submit your "Accept" recommendation.

Reviewer #2: All comments have been addressed

2. Is the manuscript technically sound, and do the data support the conclusions?

Reviewer #2: Yes

3. Has the statistical analysis been performed appropriately and rigorously? 

Reviewer #2: (No Response)

4. Have the authors made all data underlying the findings in their manuscript fully available?

Reviewer #2: (No Response)

5. Is the manuscript presented in an intelligible fashion and written in standard English?

Reviewer #2: (No Response)

6. Review Comments to the Author

Reviewer #2: Accepted. The author has worked on the recommended comments.

Thus, my suggestion would be to accept the paper.

7. PLOS authors have the option to publish the peer review history of their article (what does this mean?). If published, this will include your full peer review and any attached files.

Reviewer #2: No

---

## [Editor Report · Acceptance letter]

6 Mar 2023

PONE-D-22-35274R1 

Contextualizing gender disparities in online teaching evaluations for professors 

Dear Dr. Ni:

I'm pleased to inform you that your manuscript has been deemed suitable for publication in PLOS ONE. Congratulations! Your manuscript is now with our production department. 

Kind regards, 

on behalf of

Dr José Manuel Santos Jaén 

Academic Editor

PLOS ONE